# Antimicrobial Activity of Silver Camphorimine Complexes against *Candida* Strains

**DOI:** 10.3390/antibiotics8030144

**Published:** 2019-09-10

**Authors:** Joana P. Costa, M. Joana F. Pinheiro, Sílvia A. Sousa, Ana M. Botelho do Rego, Fernanda Marques, M. Conceição Oliveira, Jorge H. Leitão, Nuno P. Mira, M. Fernanda N. N. Carvalho

**Affiliations:** 1Centro de Química Estrutural, Instituto Superior Técnico, Universidade de Lisboa, Av. Rovisco Pais, 1049-001 Lisboa, Portugal; 2Instituto de Bioengenharia e Biociências, Departamento de Bioengenharia, Instituto Superior Técnico Universidade de Lisboa, Av. Rovisco Pais, 1049-001 Lisboa, Portugal; 3Centro de Química-Física Molecular e BSIRG, Instituto de Bioengenharia e Biociências, Instituto Superior Técnico Universidade de Lisboa, Av. Rovisco Pais, 1049-001 Lisboa, Portugal; 4C2TN Centro de Ciências e Tecnologias Nucleares (C2TN), Instituto Superior Técnico, Universidade de Lisboa, Estrada Nacional 10 (km 139,7), 2695-066 Bobadela LRS, Portugal

**Keywords:** silver complexes, camphorimine, anti-*Candida* activity, antifungals, antibacterials

## Abstract

Hydroxide [Ag(OH)L] (L = ^IV^L, ^V^L, ^VI^L, ^VII^L), oxide [{AgL}_2_}(μ-O)] (L = ^I^L, ^II^L, ^III^L, ^V^L, ^VI^L) or chloride [Ag^II^L]Cl, [Ag(^VI^L)_2_]Cl complexes were obtained from reactions of mono- or bicamphorimine derivatives with Ag(OAc) or AgCl. The new complexes were characterized by spectroscopic (NMR, FTIR) and elemental analysis. X-ray photoelectron spectroscopy (XPS), ESI mass spectra and conductivity measurements were undertaken to corroborate formulations. The antimicrobial activity of complexes and some ligands were evaluated towards *Candida albicans* and *Candida glabrata*, and strains of the bacterial species *Escherichia coli*, *Burkholderia contaminans*, *Pseudomonas aeruginosa* and *Staphylococcus aureus* based on the Minimum Inhibitory Concentrations (MIC). Complexes displayed very high activity against the *Candida* species studied with the lowest MIC values (3.9 µg/mL) being observed for complexes **9** and **10A** against *C. albicans*. A significant feature of these redesigned complexes is their ability to sensitize *C. albicans*, a trait that was not found for the previously investigated [Ag(NO_3_)L] complexes. The MIC values of the complexes towards bacteria were in the range of those of [Ag(NO_3_)L] and well above those of the precursors Ag(OAc) or AgCl. The activity of the complexes towards normal fibroblasts V79 was evaluated by the MTT (3-[4,5-dimethylthiazol-2-yl]-2,5 diphenyl tetrazolium bromide) assay. Results showed that the complexes have a significant cytotoxicity.

## 1. Introduction

The resistance of microorganisms to conventional antimicrobials is presently a serious threat to public health worldwide, representing a huge financial burden for public health systems. The group of bacterial pathogens known as ESKAPE is of particular concern, which includes *Enterococcus faecium*, *Staphylococcus aureus*, *Klebsiella pneumoniae*, *Acinetobacter baumannii*, *Pseudomonas aeruginosa*, and *Enterobacter* spp. [1]. In addition, fungal infections, and notably candidiasis caused by members of the *Candida* genus, are also of increasing concern worldwide. These infections range from superficial infections to life-threatening disseminated mycoses [2,3]. Although *Candida albicans* remains the major causative agent of candidiasis, there is an increase of the incidence of disseminated infections caused by *C. glabrata*, together with an increased resistance to antifungals among clinical isolates of this species [4,5].

The increasing resistance of pathogenic microorganisms to existing antimicrobials has been accompanied by a scarce increment in the number of alternative compounds commercially available, due to the low interest shown by the pharma industry in the search and development of new antimicrobials [6,7]. The present situation is an explosive combination of increasing resistance to antimicrobials and the lack of investment in novel antimicrobials. Therefore, it is urgent to find novel molecules with chemical characteristics different from those commercially available. Ideally, these molecules should display new modes of action and point to new microbial targets [8]. Aware of such a need, the scientific community has engaged in a search for novel antifungals alternative to azoles, echinocandins, or polyenes [9], as well as for novel antibacterials as alternative to penicillins, cephalosporin, tetracyclines, macrolides, quinolones or sulphonamides [10,11]. Consequently, many new molecules with antimicrobial activity have been described, including peptides from a multitude of species [12,13], natural extracts from plants, herbs, and spices [14], polymers modified with antimicrobial functional groups [15], and metal-based molecules [16].

Complexes are feasible alternatives to the most used organic compounds, since the specific properties of the metal site introduces steric and electronic characteristics relevant to switch distinct mechanisms of action (e.g. electron transfer and redox processes) [17].

The design of complexes to tailor efficient antimicrobial agents requires that the metal and the ligands are chosen according to the antifungal or antibacterial purpose, since the activity towards fungi or bacteria is commonly different [18]. The choice of silver and copper as precursors for the synthesis of complexes is attractive as these metals have been in use for thousands of years. For instance, silver and copper vessels have been used for water and food preservation since the Persian kings due to their antimicrobial properties [16].

Silver-based camphorimine complexes [Ag(NO_3_)L] emerged among the newly developed molecules with strong antimicrobial potential, having excellent antifungal activity against several pathogenic species of the *Candida* genus [18]. Despite the demonstrated efficacy of the developed camphorimine complexes in inhibiting growth of *C. glabrata*, *C. tropicalis,* and *C. parapsilosis*, there was no inhibition against *C. albicans* by silver nitrate camphorimine complexes [18], a drawback that has now been reported to be surpassed by redesign of the complexes. The characteristics of the camphor ligands to tune the properties, reactivity, and applications of the complexes was evidenced in former work [19,20,21,22]. Therefore, a new set of camphorimine complexes was synthesized that exhibits very high activity against *C. albicans*, highlighting the relevance of both the ligands (camphor derivatives) and the co-ligands (NO_3_^−^, OH^−^ or O^2−^) to achieve microbial growth inhibition.

## 2. Results

A new set of camphorimine silver complexes were synthesized using silver acetate (AgOAc) or silver chloride (AgCl) as metal precursors to tune the properties and the antimicrobial activity of the complexes. The objective was to keep the structure of the complexes while replacing the nitrate ion by a less acidic anionic co-ligand (OAc^−^ or Cl^−^), aiming to overcome the lack of antifungal activity displayed by the nitrate complexes [Ag(NO_3_)L] towards *C. albicans*. The absence of antifungal activity was accompanied by formation of silver nanoparticles (AgNPs) [18]. The acidic character of the nitrate ion was hypothesized to favor the reduction processes mediated by a protein existing in *C. albicans*, but not at other *Candida* species. Less acidic co-ligands (OAc^−^, Cl^−^) are expected to be less efficient in the activation of AgNPs formation.

The low solubility of silver acetate (AgOAc) in solvents other than water requires that reactions with camphor compounds are carried out in H_2_O/EtOH since the camphor derivatives used as ligands are not soluble in H_2_O. Solutions of silver acetate in water have acidic character (pH ca. 4), consistent with release of acetic acid (pKa = 4.76) [23] and formation of silver hydroxide (AgOH) or silver oxide (Ag_2_O) solutions (Scheme 1).

Since different forms of silver species co-exist in solution, complexes with different metal cores can be obtained, depending on the characteristics of the camphorimine derivatives (Figure 1) and the experimental conditions.

Addition of the camphorimine (OC_10_H_14_NY, Figure 1a) or bicamphorimine compounds (OC_10_H_14_N)_2_Z, Figure 1b) to the silver acetate solution increases the pH value to ca. 7–8, prompting the formation of hydroxide or oxide silver complexes. The camphor mono imine ligands (^I^L-^III^L) favor the formation of binuclear complexes with the two silver sites bridged by oxygen [{AgL}_2_(μ-O)] (**1**–**3**), while ^IV^L and the bicamphor ligands (^V^L–^VII^L) prompt the formation of hydroxide type complexes [AgL(OH)] (**4**–**6**,**10**) (Scheme 2).

The absence of OH stretches in the IR spectra of complexes **1**–**3** (Table 1) support their formulation as oxide rather than hydroxide complexes. The hydroxide complexes (**4**,**5**,**7**) fit in a 1:1 ligand to metal ratio, consistent with a coordination polymer character. At complex **4** ([Ag(^IV^L´)(OH)]) the ligand (Y=NH_2_) is protonated (^IV^L´ = ^IV^L·HOOCCH_3_), as confirmed by elemental analysis and FTIR (Table 1) through bands at 3455 and 3340 cm^−1^ (attributed to the OH^−^ and NH_4_^+^ groups) and at 1593, 1567 cm^−1^ (attributed to the acetate (COO^−^) group). The metal to ligand ratio at **6** (3:2) differs from that of all the other complexes, conceivably due to steric demands of the bicamphorimine ligand. All complexes (**1**–**7**) were characterized by spectroscopic (FTIR, NMR) and analytical techniques. The relevant spectroscopic characteristics are highlighted in Table 1 (see experimental section for further details). Complex **1** is not sufficiently soluble to obtain NMR data, thus formulation was supported by X-ray photoelectron spectroscopy (XPS).

### 2.1. Analysis of Complex ***1*** by XPS

Complex **1** was characterized by XPS. Besides the survey spectrum (not shown), the detailed regions C 1s, N 1s, O 1s, and Ag 3d were analyzed and are shown in Figure 2.

Ag 3d region displays a doublet with the main component, Ag 3d5/2, centered at 368.4 ± 0.2 eV and the minor component, Ag 3d3/2, at a BE 6 eV higher. N 1s is fittable with a single peak centered at 399.9 ± 0.2 eV. C 1s was fitted with a main peak (used to correct all the binding energies for charge accumulation effects) assigned to all the carbons just bound to other carbon atoms and/or hydrogen atoms set at 285 eV. Other peaks at 285.6 ± 0.2, 286.6 ± 0.2, and 287.1 ± 0.2 eV are assigned to carbon in C–N, in C–O and in C=O bonds, respectively. Finally, a peak at 288.8 ± 0.2 eV is assigned to the urethane group (NHCOOMe) in the ligand and to the acetate group in the precursor. Quantitative results are compatible with the coexistence of the complex and small amounts of the precursor (silver acetate). Discounting the precursor contributions, computed atomic ratios for the complex give the following values: Ag/N = 0.50, Ag/O = 0.29 and O/N = 1.75, fully consistent with the formulation [{Ag^I^L}2(µ-O)] (L = C_10_H_18_N_2_O_3_) proposed for **1**.

### 2.2. Silver Chloride Derived Complexes

The solubility of silver chloride in common solvents is even lower than that of silver acetate, however it is reasonably soluble in ammonia. Thus, the reactions of AgCl with the camphorimine derivatives (L) were performed in NH_3_·H_2_O/EtOH. In such basic medium, silver oxide exists in solution, thus accounting for the formation of the oxide complexes (**3**, **9**, **10A**). At [{Ag(NH_3_)}_2_(μ-^V^L)(μ-O)] (**9**), ammonia (NH_3_) further acts as a co-ligand. Complex **3** is either obtained from reaction of ligand ^III^L with AgCl or Ag(OAc), in agreement with the formation of silver oxides either from silver acetate or silver chloride under the experimental conditions used. By strict control of the order of addition of AgCl to the solutions of ligands ^II^L and ^VI^L, non-oxide complexes [Ag(^II^L)]Cl (**8**) and [Ag(^VI^L)_2_]Cl (**10**) were obtained. The cationic character of complex **10** was achieved by ESI-MS analysis. The ESI(+)/MS spectrum for a solution of [Ag(^VI^L)_2_]Cl shows a group of peaks at m/z 915/317 (Figure 3), consistent with the ionic complex [Ag(C_26_H_32_N_2_O_2_)_2_]^+^ formed by two neutral ligands and a silver cation, with the characteristic isotopic distribution of silver-containing species (Figure 3, right upper insert).

The cationic character of **8** was further confirmed through conductivity measurement (138 Ω^−1^.cm^2^.mole^−1^) in acetonitrile. The value is within the range (120–160 Ω^−1^.cm^2^.mole^−1^) expected for a 1:1 electrolyte [24]. Such as for the above complexes **1**–**7**, the characterization of complexes **8**–**10A** was achieved by elemental analysis, FTIR and NMR. Some relevant spectroscopic details are displayed in Table 2.

### 2.3. Antimicrobial Activity Assessment

The antimicrobial properties of the above complexes were assessed for *C. albicans* and *C. glabrata*, as well as for the bacterial pathogens *E. coli* ATCC25922, *S. aureus* Newman, *B. contaminans* IST408 and *P. aeruginosa* 477, based on the evaluation of the values of Minimum Inhibitory Concentration (MIC). The selected bacterial strains represent pathogens of medical relevance, difficult to treat and eradicate worldwide, mainly due to their resistance to multiple antibiotics. *B. contaminans* IST408 was isolated from a Portuguese Cystic Fibrosis patient [24]. *P. aeruginosa* and *S. aureus* are members of the ESKAPE group, responsible for many hospital- and community-acquired infections [1]. *E. coli* ATCC25922 is a commonly used reference in antimicrobial activity assays. The antimicrobial activities of complexes **1**, **4**, and **5** were not assessed due to their low solubility (**1**) or stability (**4**, **5**). 

All the complexes essayed display anti-*Candida* activity that range from 15.6 µg/mL (**1**, **2**, **6**, **9**, **10A**) to 125–250 µg/mL (**8**, **10**). The antibacterial activity of the complexes (Table 3) ranged from 19 µg/mL (*P. aeruginosa*, **2**) to ≥112 µg/mL (**6**, **8**, **10**). The MIC values measured for the ligands (^II^L, ^III^L, ^V^L and ^VI^L) display very high MIC values (≥500 µg/mL) consistent with their lack of antimicrobial activity.

The MIC values (Table 3) show that the complexes are active against bacterial strains *E. coli* ATCC25922, *B. contaminans* IST408, *P. aeruginosa* 477, and *S. aureus* Newman, although at relatively high MIC values. In contrast, the MIC_50_ values obtained for the two *Candida* species are very low, thereby showing that the complexes have higher antifungal than antibacterial activity. Although previous work showed that silver camphorimine complexes [Ag(NO_3_)L] have high anti *Candida* spp. activity (MIC_50_ 2.0–15.6 µg/mL for *C. parapsilosis*) the complexes **2**, **6**, **9**, **10A**, display values (7.8 µg/mL, **9**) that are even lower than those formerly reported against *C. glabrata* (MIC_50_ ≥15.6 µg/mL) [18]. More important, all complexes (except **8**) are efficient against *C. albicans*, a feature not observed for the nitrate complexes [Ag(NO_3_)L] [18]. So, by replacing nitrate by hydroxide or oxide co-ligands, we were able to synthesize complexes that inhibit growth of *C. albicans* even more efficiently (MIC_50_ 3.9 µg/mL; **9**, **10A**) than *C. glabrata*. The lack of activity of [Ag(^II^L)]Cl (**8**) towards the two *Candida* species and the bacterial strains under study is attributed to its cationic character. Complex **10**, which is also cationic, displays a relatively low activity (Table 3). These results reinforce those previously obtained for the cationic [Ag(OC_10_H_13_NOH)_2_]NO_3_ [19]. The ionic character of the complexes decreases their lipophilicity and conceivably makes it more difficult for their penetration into the intracellular environment of *Candida* spp. or bacterial cells, thereby reducing their activity.

In general, the complexes display both antibacterial and antifungal activities, although they perform better as antifungals than as antibacterials. The MIC values show that some of the complexes have very high antibacterial activity (40–60 µg/mL) and/or excellent antifungal activities (4–8 µg/mL). Overall, the complexes perform better for Gram-negative (MIC 19–61 µg/mL) than for Gram-positive bacteria (*S. aureus* Newman, 58–250 µg/mL). Complexes [{Ag(^II^L)}_2_(µ-O)] (**2**) and [{Ag(NH_3_)}_2_(μ-^V^L)(μ-O)] (9**)** display the highest activity against *P. aeruginosa* 477 (MIC, 19 µg/mL) while complex [Ag_2_(μ-^VI^L)(µ-O)] (**10A**) displays the highest activity towards *B. contaminans* IST408 (MIC 23 ± 3 µg/mL). These complexes have in common a dinuclear character with the two metals sharing an oxygen atom and camphorimine ligands that may prompt electron delocalization through the aromatic ring. Such characteristics conceivably are not just circumstantial for their antimicrobial activity.

To obtain insights into the toxicity of the complexes towards mammalian cells, the IC_50_ measurements of representative complexes were evaluated towards V79 normal fibroblasts which are cells commonly used to assess the toxicological effects of drugs [25]. Data shows that IC_50_ values of the complexes are low and comparable or even lower than those of the MIC values obtained for fungi (Table 3).

These results were not completely unexpected since fungi and mammalian cells are eukaryotes and therefore some of the mechanisms by which the complexes exert toxicity against the *Candida* spp. may be conserved in the mammalian cells [26]. This difficulty in achieving specificity is a recognized challenge in the design of new compounds selectively targeting fungal cells. Future work will focus on the design of silver camphorimine complexes with both reduced cytotoxicity and enhanced antimicrobial activities.

## 3. Materials and Methods

### 3.1. General Procedures

The camphorimines were obtained from camphorquinone by reaction with the appropriate amine or hydrazine in ethanol using reported procedures [18,27,28,29]. In the case of air sensitive complexes, Schlenk and vacuum techniques were used. Ethanol was purchased from Fisher Scientific, ammonia from Sigma-Aldrich and acetonitrile from Carlo Erba. The amines and hydrazines were purchased from Sigma-Aldrich and silver acetate and silver chloride from Merck.

The IR spectra were obtained from KBr pellets using a JASCO FT/IR 4100 spectrometer. The NMR spectra (^1^H, ^13^C, DEPT, HSQC and HMBC) were obtained from MeOH-d4, CD_3_CN or CD_2_Cl_2_ solutions using Bruker Avance II+ spectrometers (300 or 400 MHz). NMR chemical shifts are referred to tetramethylsilane (TMS) (δ = 0 ppm).

The ESI mass spectrum was obtained on a LCQFleet ion trap mass spectrometer equipped with an electrospray source (Thermo Scientific^TM^, Waltham, MA USA), operating in the positive ion mode. The XPS data was obtained using a Kratos XSAM800 equipment.

Conductivity was measured in acetonitrile (1.0 × 10^−3^ M solution) at 25 °C using a CON 510 bench conductivity meter provided with a Conductivity/TDS electrode (code No. ECCONSEN91W/ 35608-50, K = 1).

### 3.2. Synthesis

Complexes **1**–**7** were obtained from reaction of the suitable ligand with silver acetate (1:1). Air was partially excluded by bubbling of nitrogen (3 minutes). The reaction mixtures were protected from light to preclude reduction of Ag^+^ to Ag^0^. The typical procedure is described for **1**. Complexes **8**–**10** were obtained from reaction of the suitable ligand with silver chloride (1:1). A typical procedure is described for **8**.

[{Ag^I^L}_2_(µ-O)] (**1**)—A solution of the camphorimine OC_10_H_14_NNHCOOMe (**^I^**L, 72 mg, 0.35 mmol**)** in EtOH (5 mL) was added to a suspension of Ag(OAc) (50 mg, 0.35 mmol) in H_2_O (5 mL) under N_2_. The whitish suspension was stirred for ca. 1h. The slight suspension still remaining was then removed by filtration and the solution was evaporated to dryness to yield the complex. Yield 55%. Elem. Anal. (%) for Ag_2_C_24_H_36_N_4_O_7_. Found: C, 41.6; N, 7.9; H, 5.1; Calc.: C, 41.4; N, 7.5; H, 5.5. IR (cm^−1^): 1722 (C=O), 1649 (C=N), 1588 (O=COMe). ^1^H NMR (400 MHz, MeOH-d4, δ ppm): 4.56 (s, 3H), 3.81 (sbr, 1H), 1.88–1.81 (m, 2H), 1.54–1.44 (m, 2H), 1.18 (s, 3H), 1.04 (s, 3H), 0.85 (s, 3H). ^13^C NMR Decomposes during acquisition.

[{Ag^II^L}_2_(μ-O)] (**2**)—OC_10_H_14_NC_6_H_4_ (**^II^L**, 72 mg; 0.3 mmol) in ethanol (5 mL) and silver acetate (50 mg; 0.3 mmol) in H_2_O (5 mL) were stirred for 4 h. Yield 74%. Elem. Anal. (%) for Ag_2_C_32_H_36_N_2_O_3_. Found: C, 54.9 N, 3.5; H, 5.8; Calc.: C, 54.7; N, 3.8; H, 6.2. IR (cm^−1^): 1745 (C=O), 1652 (C=N), 1567 (CH_arom_). ^1^H NMR (300 MHz, CD_3_CN, δ ppm) 7.40 (t, *J* = 7.8 Hz, 2H), 7.19 (t, *J* = 7.5 Hz, 1H), 6.90 (d, *J* = 7.5 Hz, 2H), 2.74 (d, *J* = 5.1 Hz, 1H), 1.90 (mc, 2H), 1.61 (mc, 2H) 1.04 (s, 3H), 0.97 (s, 3H), 0.86 (s, 3H). ^13^C NMR (300 MHz, CD_3_CN, δ ppm): 207.4, 173.3, 130.1, 126.0, 120.9, 59.0, 51.0, 45.3, 30.8, 24.8, 21.16, 17.6, 9.4.

[{Ag^III^L}_2_(μ-O)] (**3**)—OC_10_H_14_NC_6_H_4_CH_3_-4 (^III^L, 76 mg; 0.3 mmol) in ethanol (5 mL) and silver acetate (50 mg; 0.3 mmol) in H_2_O (5 mL) were stirred for 4 h. Yield 89%. Elem. Anal. (%) for Ag_2_C_34_H_42_N_2_O_3_. Found: C, 55.3 N, 3.5; H, 5.9; Calc.: C, 55.0; N, 3.8; H, 5.7. IR (cm^−1^): 1747 (C=O), 1653 (C=N), 1565 (CH_arom_). ^1^H NMR (300 MHz, CD_2_Cl_2_, δ ppm): 7.24 (d, *J* = 7.5 Hz, 2H), 6.97 (d, *J* = 7.5 Hz, 2H), 2.92 (d, *J* = 4.8 Hz, 1H), 2.36 (s, 3H), 2.3–1.9 (m, 2H), 1.6–1.5 (m, 2H), 1.61 (s, 3H) 1.07 (s, 3H), 0.98 (s, 3H), 0.87 (s, 3H). ^13^C NMR (300 MHz, CD_2_Cl_2_, δ ppm): 206.9, 174.2, 146.2, 138.6, 130.9, 123.2, 59.1, 52.2, 46.0, 31.3, 24.9, 21.3, 21.0, 17.4, 9.2. 

Compound **3** can alternatively be obtained from reaction of AgCl (50 mg; 0.35 mmol in 5 mL of ammonia 33%) with ^III^L (89 mg; 0.35mmol in 5 mL EtOH) upon stirring overnight, filtration to eliminate residues of silver followed by solvent evaporation until precipitation which is then dried under vacuum to obtain 3. The yield (60%) is lower than that in reaction of ^III^L with Ag(OAc).

[Ag(^IV^L)(OH)]·CH_3_COOH (**4**)—OC_10_H_14_NC_6_H_4_NH_2_-4 (^IV^L, 46 mg; 0.18 mmol) in ethanol (3 mL) and silver acetate (30 mg; 0.18 mmol) in H_2_O (3 mL). The solutions were degassed. The mixture was stirred for 4 h under N_2_. Yield 57%. Elem. Anal. (%) for AgC_18_H_25_N_2_O_4_. Found: C, 49.3; N, 6.2; H, 5.4; Calc.: C, 49.0; N, 6.4; H, 5.7; IR (cm^−1^): 3454 (OH), 3339 (NH_2_), 1733 (C=O), 1625 (C=N), 1593 (CH_arom_), 1567 (O=CO), 1504 (NH_2_). ^1^H NMR (300 MHz, CD_3_CN, δ ppm): 6.93 (d, *J* = 8.6 Hz, 2H), 6.70 (d, *J* = 8.6 Hz, 2H), 4.30 (sbr, 2H) 2.98 (d, *J* = 4.7 Hz, 1H), 1–91–1.84 (m, 3H), 1.68–1.51 (m, 3H), 1.06 (s, 3H), 1.00 (s, 3H), 0.83 (s, 3H). ^13^C NMR (300 MHz, CD_3_CN, δ ppm): 207.0, 169.1, 161.3, 146.1, 139.8, 124.5, 115.3, 58.0, 51.0, 45.3, 30.8, 24.4, 20.9, 17.7, 9.2.

[{Ag(OH)(^V^L)] (**5**)—A solution of 4-C_6_H_4_(OC_10_H_14_N)_2_ (**^V^**L, 117 mg; 0.3 mmol) in ethanol (5 mL) and silver acetate (50 mg; 0.3 mmol) in H_2_O (5 mL) were stirred for 5 h. The complex precipitates from reaction mixture. Yield 54%. Elem. Anal. (%) for AgC_26_H_33_N_2_O_3_·H_2_O. Found: C, 57.2 N, 4.8; H, 6.1; Calc.: C, 57.0; N, 5.1; H, 6.4. IR (cm^−1^) 1750 (C=O), 1649 (C=N), 1559 (CH_arom_). ^1^H NMR(300 MHz, CD_3_CN, δ ppm) 6.98 (s, 4H), 2.86 (d, *J* = 4.9 Hz, 2H), 1.9 (mc, 3H), 1.05 (s, 6H), 0.99 (s, 6H), 0.86 (s, 6H). ^13^C NMR (300 MHz, CD_3_CN, δ ppm): 207.3, 173.2, 148.0, 122.4, 58.9, 51.2, 45.3, 30.9, 24.9, 21.2, 17.6, 9.4.

[{Ag(OH)}_3_(^VI^L)_2_] (**6**)—3-C_6_H_4_(OC_10_H_14_N)_2_ (**^VI^L**, 199 mg; 0.5 mmol) in ethanol (8.5 mL) was added to a solution of silver acetate (95 mg; 0.5 mmol) in H_2_O (8.5 mL) and the mixture stirred overnight. Yield 62%. Elem. Anal. (%) for Ag_3_C_52_H_67_N_4_O_7_ Found: C, 53.1 N, 4.3; H, 5.6; Calc.: C, 52.8; N, 4.7; H, 5.7. IR (cm^−1^): 1751 (C=O), 1668 (C=N), 1566 (CH_arom_). ^1^H NMR (400 MHz, CD_2_Cl_2_, δ ppm): 7.34 (t, *J* = 7.9, Hz,1H), 6.68 (2d, *J* = 1.9 Hz, 2H), 6.34 (s, 1H), 2.77 (d, *J* = 4.4 Hz, 2H), 2.16–2.01 (m, 2H), 1.94–1.80 (m, 2H), 1.68–1.54 (m, 4H), 1.03 (s, 6H), 0.94 (s, 6H), 0.85 (s, 6H). ^13^C NMR(400 MHz, CD_2_Cl_2_, δ ppm): 206.6, 173.0, 151.1, 130.1, 117.2, 111.4, 58.5, 50.6, 44.8, 30.5, 24.7, 21.1, 17.6, 9.2.

[Ag(OH)(^VII^L)] (**7**)—**^VII^L** (44 mg; 0.3 mmol) in ethanol (10 mL) was added to Ag(OAc) (100 mg; 0.6 mmol) in H_2_O (10 mL) and stirred overnight. Yield 88%. Elem. Anal. (%) for AgC_32_H_37_N_2_O_3_ Found: C, 64.0; N, 4.3; H, 6.1; Calc.: C, 63.6; N, 4.6; H, 6.2. IR (cm^−1^): 3421 (OH), 1745 (C=O), 1660 (C=N), 1566 (CH_arom_). ^1^H NMR (300 MHz, CD_2_Cl_2_, δ ppm): 7.65 (d, *J* = 8.1 Hz, 4H), 7.03 (d, *J* = 7.9 Hz, 4H), 2.89 (d, *J* = 4.5 Hz, 2H), 2.21–2.07 (m, 2H), 1.96–1.81 (m, 2H), 1.72–1.60 (m, 4H), 1.10 (s, 6H), 1.01 (s, 6H), 0.92 (s, 6H). ^13^C NMR (300 MHz, CD_2_Cl_2_, δ ppm): 206.7, 172.5, 149.3, 137.9, 127.8, 121.5, 58.4, 50.7, 44.9, 30.6, 24.7, 21.1, 17.7, 9.2.

[{AgCl(^II^L)] (**8**)—To a solution of AgCl (50 mg; 0.35 mmol) in ammonia (33%,5 mL) a solution of the ligand ^III^L (84 mg; 0.35 mmol) in EtOH (5 mL) was added. N_2_ was then bubbled for a few minutes. The mixture was stirred for 3 h at RT. The slight suspension was filtered off and the volume of the solution was reduced until precipitation. Upon filtration the title compound was obtained and dried under vacuum. Yield 59%. Elem. Anal. (%) for AgClC_16_H_19_NO. Found: C, 50.3; N, 3.6; H, 5.1; Calc.: C, 50.0; N, 3.6; H, 5.0. IR (cm^−1^): 1744 (CO), 1651 (CN), 1589 (CH_arom_). ^1^H NMR (400 MHzCD_3_CN, δ ppm): 7.40 (t, *J* = 7.5 Hz, 2H), 7.19 (t, *J* = 7.3 Hz 1H), 6.90 (d, *J* = 7.7 Hz, 2H), 2.74 (d, *J* = 4.3 Hz, 1H), 1.91–1.84 (m, 2H), 1.66–1.55 (m, 2H) 1.04 (s, 3H), 0.97 (s, 3H), 0.86 (s, 3H). ^13^C NMR(400 MHz, CD_3_CN, δ ppm): 207.5, 150.9, 130.1, 126.0, 120.9, 58.9, 51.1, 45.2, 30.8, 24.8, 21.2, 17.6, 9.4.

[{Ag(NH_3_)}_2_(μ-^V^L)(μ-O)] (**9**)—A solution of the ligand ^V^L (75 mg; 0.19 mmol) in EtOH (2.5 mL) was added to a solution of AgCl (23 mg; 0.16 mmol) in ammonia (33%, 2.5 mL). The mixture was stirred for 4 h at RT. Yield 69%. Anal. (%) for Ag_2_C_26_H_38_N_4_O_3_. Found: C, 46.4 N, 8.0; H, 5.3; Calc.: C, 46.6; N, 8.4; H, 5.7. IR (cm^−1^): 3341 (NH), 1753 (CO), 1621 (CN), 1595 (CH_arom_), 1505 (NH). ^1^H NMR (400 MHz, MeOH-d_4_, δ ppm): 7.06 (s, 4H), 2.92 (d, *J* = 4.4 Hz, 2H), 2.29–2.17 (m, 2H), 2.02–1.93 (m, 2H), 1.75–1.56 (m, 4H) 1.09 (s, 6H), 1.07 (s, 6H), 0.83 (s, 6H). ^13^C NMR (400 MHz, MeOH-d_4,_ δ ppm): 190.5, 169.8, 148.8, 125.4, 123.1, 116.2, 58.9, 49.5, 45.7, 31.3, 25.0, 21.3, 17.9, 9.3. There is evidence for two isomers in solution that were not further investigated.

[AgCl(^VI^L)_2_] (**10**)—Solid AgCl (50 mg; 0.35 mmol) was added to ^VI^L (137 mg; 0.35 mmol) in Et_2_OH (5 mL) followed by addition of ammonia (33%, 5 mL). The mixture was stirred for 3 h. Yield 72%. Anal. (%) AgClC_52_H_64_N_4_O_4_. Found: C, 65.9 N, 5.8; H, 6.8; Calc.: C, 65.6; N, 5.9; H, 6.8. IR (cm^−1^): 1750 (CO), 1661 (CN), 1579 (CH_arom_). ^1^H NMR (400 MHz, CD_3_CN, δ ppm): 7.4 (t, *J* = 7.9 Hz, 1H); 6.7 (2d, *J* = 7.9 Hz, 2H), 6.34 (m, 1H), 2.77 (d, *J* = 4.5 Hz, 2H), 1.04 (s, 3H), 0.98 (s, 3H), 0.87 (s, 3H). ^13^C NMR (400 MHz, CD_3_CN, δ ppm): 207.2, 173.8, 152.1, 131.1, 111.3, 59.0, 51.2, 45.2, 30.8, 24.9, 21.2, 17.6, 9.4.

[Ag_2_(μ-^VI^L)(µ-O)] (**10A**)—Addition of ^VI^L (137 mg; 0.35 mmol, 5 mL H_2_O) after complete dissolution of AgCl (50 mg; 0.35 mmol) in ammonia (33%, 5 mL) followed by stirring overnight. Yield 56%. Anal. (%) Ag_2_C_26_H_32_N_2_O_3_·H_2_O. Found: C, 47.7; N, 4.5; H, 5.0. Calc.: C, 47.7; N, 4.3; H, 5.2. IR (cm^−1^): 1749 (CO), 1660 (CN), 1579 (CH_arom_). ^1^H NMR (400 MHz, CD_3_CN, δ ppm): 7.43 (t, *J* = 7.9 Hz, 1H), 6.7 (dd, *J* = 1.7, Hz *J* = 7.9 Hz, 2H), 6.34 (m, 1H), 2.77 (d, *J* = 4.9 Hz, 2H), 2.12–2.03 (m, 2H), 1.91–1.75 (m, 2H), 1.64–1.56 (m, 4H), 1.04 (s, 6H), 0.98 (s, 6H), 0.87 (s, 6H). ^13^C NMR (400 MHz, CD_3_CN, δ ppm): 207.3, 173.9, 152.0, 130.9, 117.4, 111.3, 59.0, 51.1, 45.1, 30.8, 24.9, 21.2, 17.6, 9.3.

### 3.3. Antibacterial Activity Determinations

The antibacterial activity of compounds was assessed by determining their Minimal Inhibitory Concentration (MIC) towards the Gram-positive *Staphylococcus aureus* Newman and the Gram-negative *Escherichia coli* ATCC 25922, *Pseudomonas aeruginosa* 477 and *Burkholderia contaminans* IST408. These bacterial strains are clinical isolates and were chosen as representatives of important bacterial pathogens [1,25,30,31,32]. MICs were determined using Mueller Hinton Broth (MHB; Becton, Dickinson and Company) as growth medium, based on microdilution assays, using previously described methods [20,21]. In brief, a colony from a bacterial culture freshly grown in MHB solid medium was transferred into MHB liquid medium and incubated for 4–5 h with agitation (250 rpm) at 37 °C. Bacterial cultures were then diluted using fresh MHB to obtain ca. 10^6^ colony forming units (CFUs) per mL. These cultures were used to inoculate approximately 5 × 10^5^ CFUs per mL in 96-well polystyrene microtiter plates containing 100 μL of MHB supplemented with different concentrations of each compound, obtained by 1:2 serial dilutions ranging 0.5 to 512 μg/mL. Stock solutions of compounds were prepared with DMSO. After inoculation, microtiter plates were incubated at 37 °C for 20 h. Bacterial growth was assessed by measuring the cultures optical density at 640 nm, in a SPECTROstarNano (BMG LABTECH) microplate reader. Experiments were carried out at least four times in duplicates. Wells containing 100 μL of 1× concentrated MHB and 100 μL of 10^6^ CFUs per mL were used as positive controls, while wells containing 200 μL of sterile MHB 1 × concentrated were used as negative controls.

### 3.4. Assessment of Complexes Anti-Candida Activity

The ability of the complexes 1–10A or of the ligands to inhibit growth of *C. albicans* or *C. glabrata* was assessed using the highly standardized microdilution method recommended by EUCAST (European Committee on Antimicrobial Susceptibility Testing). The MIC_50_ values were considered to be the concentration of drug that reduced yeast growth by more than 50% of the growth registered in drug-free medium [33]. The strains used in this work were *C. albicans* SC5314 and *C. glabrata* CBS138, largely used as reference strains. Briefly, cells of the different species were cultivated (at 30 °C and with 250 rpm orbital agitation) for 17 h in Yeast Potato Dextrose (YPD) growth medium and then diluted in fresh Roswell Park Memorial Institute (RPMI) growth medium (Sigma) to obtain a cell suspension having an OD_530nm_ of 0.05. From these cell suspensions, 100 µL aliquots were mixed in the 96-multiwell polystyrene plates with 100 µL of fresh RPMI medium (control) or with 100 µL of this same medium supplemented with 0.98–500 μg/mL of the different compounds. As a control we also examined the inhibitory effect of Ag(OAc) or of AgCl. After inoculation, the 96-multiwell plates were incubated without agitation at 37 °C for 24 h. After that time, cells were re-suspended and the OD_530nm_ of the cultures was measured in a SPECTROstarNano (BMG LABTECH) microplate reader. The MIC_50_ value was taken as being the highest concentration tested at which the growth of the strains was 50% of the value registered in the control lane.

### 3.5. Toxicity Assessment

The toxicity of the compounds was evaluated towards normal fibroblasts V79, obtained from the American Type Culture Collection (ATCC). The cell lines were grown in Dulbeco´s Modified Eagle Medium (DMEM) + Glutamax^®^ medium supplemented with 10% Fetal Bovine serum (FBS) and maintained in a humidified atmosphere at 37 °C using an incubator (Heraeus, Germany) with 5% CO_2_. Cell viability was measured by the MTT (3-[4,5-dimethylthiazol-2-yl]-2,5 diphenyl tetrazolium bromide) assay, based on the conversion of the tetrazolium bromide into formazan crystals by living cells which determines mitochondria activity [34]. For the assay, cells were seeded in 96-well plates at a density of 10^4^ cells per well in 200 μL medium and allowed to attach overnight. Complexes were first diluted in DMSO to solubilize and then in medium to prepare the serial dilutions in the range 10^−7^–10^−4^ M. The maximum concentration of DMSO in the medium (1%) had no toxicity effect. After careful removal of the medium, 200 μL of each dilution were added to the cells, and incubated for another 48 h at 37 °C. At the end of the treatment, the medium was aspirated and 200 μL of MTT solution (1.5 mM in PBS) was applied to each well. After 3 h at 37 °C, the medium was discarded and 200 μL of DMSO was added to solubilize the formazan crystals. The cellular viability was assessed by measuring the absorbance at 570 nm using a plate spectrophotometer (Power Wave Xs, Bio-Tek). The IC_50_ values were calculated using the GraphPad Prism software (version 5.0). Results are mean ± SD of at least two independent experiments done with six replicates each and represent the percentage of cellular viability related to the controls (no treatment).

### 3.6. X-ray Photoelectron Spectroscopy

For X-ray photoelectron spectroscopy characterization (XPS), a XSAM800 XPS dual anode spectrometer from KRATOS was used. The unmonochromatic Mg Kα radiation (main line at hν = 1256.6 eV) was used. Operating conditions, data acquisition and data treatment are described elsewhere [35]. For charge correction purposes, carbon bound to carbon and hydrogen in C 1s peak was set to a binding energy (BE) of 285 eV. For quantification purposes, the following sensitivity factors were used: 0.318 for C 1s, 0.736 for O 1s, 0.505 for N 1s, and 6.345 for Ag3d.

## 4. Conclusions

Silver hydroxide [Ag(OH)L] (L = ^IV^L, ^V^L, ^VI^L, ^VII^L), oxide [{AgL}_2_}(μ-O)] (L = ^I^L, ^II^L, ^III^L, ^V^L, ^VI^L) and homoleptic [Ag^II^L]Cl, [Ag(^VI^L)_2_]Cl camphorimine complexes were synthesized using Ag(OAc) or AgCl as metal sources. The basic characteristics of the reaction medium prompted the formation of hydroxide or oxide rather than acetate or chloride complexes. The selection of the camphorimine ligands (L) that encompass mono- and bi-camphors, allowed the design of complexes with considerable distinct electronic and steric properties and thus different biological activities.

In summary, the most relevant achievement of this study is that the new oxo and hydroxo silver camphorimine complexes overreach the resistance of *C. albicans*. Additionally, the complexes reach MIC_50_ values for *C. glabrata* even lower than those previously reported for the camphorimine nitrate complexes [Ag(NO_3_)L]. In fact, the antifungal activity of the oxo and hydroxo silver camphorimine complexes is even higher towards *C. albicans* than *C. glabrata*.

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
