# Peer review of "Antimicrobial Activity of Silver Camphorimine Complexes against Candida Strains"

_antibiotics, 2019, doi:10.3390/antibiotics8030144_

Round 1

Reviewer 1 Report

Overall, solid paper worthy of publication. The authors prepared a series of silver hydroxide and oxide camphorimine complexes and evaluated their activity against bacteria and fungi. While antibacterial activity is noted, the main conclusion is that several prepared complexes were active against C. albicans. However, the active complexes show significant cytotoxicity towards a human cell line which presents a challenge for further development. Specific items noted below:

-Line 22: XPS should be defined here as it is not a common technique like NMR or FTIR

-Line 34: cytotoxicity is mispelled

-Line 43: Sentence beginning here is hard to read and should be rewritten.

-Line 56: There is a non-English phrase that should be deleted.

-Line 56: The sentence beginning here ("The choice of silver...") is hard to read and should be rewritten.

-Line 70: "was" should be changed to "were"

-Line 73: "overcome" should be changed to "overcoming"

-Line 80: "on" should be changed to "out in"

-The numbering between Figure 1 and Scheme 2 is confusing as a reader. Fig. 1 refers to compounds using "IL" numbering, while Scheme 2 use "1" numbering. Additionally, there are seven compounds in Figure 1, but Scheme 2 shows eight. Looking at Tables later in the manuscript allows one to decipher the numbering but it is not clear and should be easier to follow.

-Line 151, 152: Make sure all compound numbers are BOLD

-In Table 3, the authors show MIC (inhibiting all growth) for both fungi and bacteria. However, the methods state for fungi an MIC50 (reducing growth by 50%) was determined. Consistency is needed.

-Line 194: there appears to be an unnecessary space

-Lines 223-230: Seems more appropriate for the results section. The main point of the paper is the antifungal activity so that should be emphasized in the conclusion section.

-Line 265: "Found.:" should be "Found:" (please correct in all synthetic procedures)

-Frequencies of each 1H and 13C NMR spectra should be given (e.g. 1H NMR (400 MHz, CD3CN, δ ppm)... 13C NMR (100 MHz, CD3CN, δ ppm)...)

-Coupling constants should be given for all NMR signals (i.e. triplets also, not just doublets)

-Line 321, 332, 341, 348: There is a lack of consistency with the ammonia (33%) formatting.

-Line 386: "AgOC" should be "AgOAc"

Author Response

Manuscript antibiotics-589705

POINT BY POINT Answer to criticisms raised by reviewers

REVIEWER #1

Comment: Overall, solid paper worthy of publication. The authors prepared a series of silver hydroxide and oxide camphorimine complexes and evaluated their activity against bacteria and fungi. While antibacterial activity is noted, the main conclusion is that several prepared complexes were active against C. albicans. However, the active complexes show significant cytotoxicity towards a human cell line which presents a challenge for further development.

ANSWER: Thank you for your comments. Indeed, we have now a novel challenge which is to prepare novel compounds with reduced cytotoxicity. This will be our next task. A final sentence was added, lines 242-243, to the manuscript to accommodate this comment, which now reads as follows: “Future work will focus on the design of silver camphorimine complexes with both reduced cytotoxicity and enhanced antimicrobial activities.”

Specific items noted below:

-Line 22: XPS should be defined here as it is not a common technique like NMR or FTIR

ANSWER: The acronym XPS is now defined. New line 22 now reads as follows: “X-ray photoelectron spectroscopy (XPS), …”

-Line 34: cytotoxicity is misspelled

ANSWER: Thanks. The word is now corrected.

-Line 43: Sentence beginning here is hard to read and should be rewritten.

ANSWER:

We agree. The sentence was rewritten, and lines 51-54 now reads as follows: “The present situation is an explosive combination of increasing resistance to antimicrobials and the lack of investment in novel antimicrobials. Therefore, it is urgent to find novel molecules with chemical characteristics different from those commercially available. Ideally, these molecules should display new modes of action and point new microbial targets [].

-Line 56: There is a non-English phrase that should be deleted.

ANSWER:Thanks. The non-English phrase was deleted.

-Line 56: The sentence beginning here ("The choice of silver...") is hard to read and should be rewritten.

ANSWER: We agree. The sentence was rewritten and lines 66-69 now reads as follows: “The choice of silver and copper as precursors for the synthesis of complexes is attractive these metals are in use for thousands of years. For instance, silver and copper vessels have been used for water and food preservation since the Persian kings due to their antimicrobial properties.“

-Line 70: "was" should be changed to "were"

ANSWER: Thanks. Corrected.

-Line 73: "overcome" should be changed to "overcoming"

ANSWER: Thanks. We substituted “aiming at overcome” by “aiming to overcome”.

-Line 80: "on" should be changed to "out in".

ANSWER: Thanks. We have done the change.

-The numbering between Figure 1 and Scheme 2 is confusing as a reader. Fig. 1 refers to compounds using "IL" numbering, while Scheme 2 use "1" numbering.

Additionally, there are seven compounds in Figure 1, but Scheme 2 shows eight. Looking at Tables later in the manuscript allows one to decipher the numbering but it is not clear and should be easier to follow.

ANSWER: We agree with the comment. To overcome this difficulty in interpretation, we have modified the caption of Scheme 2. We hope this new caption for Scheme 2 clarifies this point. Scheme 2 caption now reads as follows: “Types of complexes obtained from ligands IL – VIL (see Tables 1 and 2 for details).”

-Line 151, 152: Make sure all compound numbers are BOLD

ANSWER: Thanks. Corrected.

-In Table 3, the authors show MIC (inhibiting all growth) for both fungi and bacteria. However, the methods state for fungi an MIC50 (reducing growth by 50%) was determined. Consistency is needed.

ANSWER: Thank you for this observation, our mistake.  In fact, we missed to indicate MIC50 in Table 3 for fungi. This is now corrected. Additionally, whenever referring to fungi, MIC was substituted by MIC50 throughout the text.

-Line 194: there appears to be an unnecessary space

ANSWER:  The space was deleted.

-Lines 223-230: Seems more appropriate for the results section. The main point of the paper is the antifungal activity so that should be emphasized in the conclusion section.

ANSWER: Thank you for the suggestion. We have moved this part of the text to the “Results” section. Se new lines 222-232.

-Line 265: "Found.:" should be "Found:" (please correct in all synthetic procedures)

ANSWER: Thanks. The correction was made for all the synthetic procedures.

-Frequencies of each 1H and 13C NMR spectra should be given (e.g. 1H NMR (400 MHz, CD3CN, δ ppm)... 13C NMR (100 MHz, CD3CN, δ ppm)...)

ANSWER: Thanks. The frequencies were included in the revised version.

-Coupling constants should be given for all NMR signals (i.e. triplets also, not just doublets)

ANSWER: Thanks. The constants were included in the text.

-Line 321, 332, 341, 348: There is a lack of consistency with the ammonia (33%) formatting.

ANSWER: Thanks for the comment. This was corrected.

-Line 386: "AgOC" should be "AgOAc"

ANSWER: Right. Thanks. This is now corrected.

Reviewer 2 Report

Overall the authors of this paper show that the production of camphorimine silver complexes using silver acetate and silver chloride can be effective against Candida species, specifically albicans, and to a limited degree some bacterial species. However the complexes examined also showed significant cytotoxicity with human cells. This latter point makes it hard to understand the impact of this paper if the complexes show antifungal activity but have little clinical relevance due to cytotoxicity. There was little discussion on what changes may decrease the cytotoxicity or a historical perspective of how this has been handled with previous antifungal agents. 

Additionally I feel the introduction could use more background in places. There is one sentence (Lines 56-57) about the use of silver and copper in ancient times - without further elaboration it suggests the usage of these metals is restricted to ancient times. The paper also focuses on Candida species with no mention of their clinical relevance. How prominent are the various species examined in the hospitals? Since more emphasis was placed on albicans - does it that mean it accounts for more infections? The testing of several bacteria is mentioned in the abstract but not the introduction - why are those bacteria specifically included?

Issues within text:

Line 45: change to the scientific community has engaged in a search

Line 56: Error?

Lines 56-58: sentence is awkward needs to be reworded

Line 59 - line of sentence is awkward and needs reworking

Line 61: should there be a reference after sentences?

line 73 - aiming to instead of aiming at

Author Response

REVIEWER #2

Comment: Overall the authors of this paper show that the production of camphorimine silver complexes using silver acetate and silver chloride can be effective against Candida species, specifically albicans, and to a limited degree some bacterial species. However the complexes examined also showed significant cytotoxicity with human cells. This latter point makes it hard to understand the impact of this paper if the complexes show antifungal activity but have little clinical relevance due to cytotoxicity. There was little discussion on what changes may decrease the cytotoxicity or a historical perspective of how this has been handled with previous antifungal agents. 

ANSWER: Thanks for the appreciation of our work. Regarding your last comment, we added a final sentence to the “Results” section, lines 242-243, which now reads as follows: Future work will focus on the design of silver camphorimine complexes with both reduced cytotoxicity and enhanced antimicrobial activities.”

Comment: Additionally I feel the introduction could use more background in places. There is one sentence (Lines 56-57) about the use of silver and copper in ancient times - without further elaboration it suggests the usage of these metals is restricted to ancient times.

ANSWER: Thanks for this comment, with which we agree. The phrase is now re-written and new lines 66-69 now reads as follows: “The choice of silver and copper as precursors for the synthesis of complexes is attractive these metals are in use for thousands of years. For instance, silver and copper vessels have been used for water and food preservation since the Persian kings due to their antimicrobial properties [].“

Comment: The paper also focuses on Candida species with no mention of their clinical relevance. How prominent are the various species examined in the hospitals? Since more emphasis was placed on albicans - does it that mean it accounts for more infections?

ANSWER: Thanks for this comment. We have included additional information in the Introduction section. New lines 42-47 now reads as follows: In addition, fungal infections, and in particular candidiasis caused by members of the Candida genus, are also of increasing concern worldwide. These infections range from superficial infections to life-threatening disseminated mycoses [2, 3]. Although Candida albicans remains the major causative agent of candidiasis, there is an increase of the incidence of disseminated infections caused by C. glabrata, together with an increased resistance to antifungals among clinical isolates of this species [4, 5].

Comment: The testing of several bacteria is mentioned in the abstract but not the introduction - why are those bacteria specifically included?

ANSWER: Thanks for the observation. We added the following sentence to the “Introduction” section to accommodate this criticism (see new lines 40-42): “The group of bacterial pathogens known as ESKAPE is of particular concern, which includes Enterococcus faecium, Staphylococcus aureus, Klebsiella pneumoniae, Acinetobacter baumannii, Pseudomonas aeruginosa, and Enterobacter spp. [1].

 In addition, in the “Results section a brief justification for the specific bacterial species chosen was added, new lines 189-193: The selected bacterial strains represent pathogens of medical relevance, difficult to treat and eradicate worldwide, mainly due to their resistance to multiple antibiotics. B. contaminans IST408 was isolated from a Portuguese cystic fibrosis patient [24]. P. aeruginosa and S. aureus are members of the ESKAPE group, responsible for many hospital- and community-acquired infections [1]. E. coli ATCC25922 is a commonly used reference in antimicrobial activity assays.

Issues within text:

Line 45: change to the scientific community has engaged in a search

ANSWER: Thanks for the suggestion. This correction was made.

Line 56: Error?

ANSWER: This sentence was deleted.

Lines 56-58: sentence is awkward needs to be reworded

ANSWER: We agree. The sentence was rewritten, new lines 66-69, and now reads as follows: “The choice of silver and copper as precursors for the synthesis of complexes is attractive these metals are in use for thousands of years. For instance, silver and copper vessels have been used for water and food preservation since the Persian kings due to their antimicrobial properties [].“

Line 59 - line of sentence is awkward and needs reworking

ANSWER: We agree. The sentence was re-written, new lines 70-72, which now reads as follows: Silver-based camphorimine complexes [Ag(NO3)L] emerged among the newly developed molecules with strong antimicrobial potential as having excellent antifungal activity against several pathogenic species of the Candida genus [18].

Line 61: should there be a reference after sentences?

ANSWER: The reference was included at the end of the sentence ending in line 72.

line 73 - aiming to instead of aiming at

ANSWER: Corrected, as requested. See new line 85.
